# Quality of life among adults with scabies: A community-based cross-sectional study in north-western Ethiopia

**Robel Yirgu**[1,2,3]*, **Jo Middleton**[1,4], **Jackie A. Cassell**[1,4], **Stephen Bremner**[1,4], **Gail Davey**[1,2], **Abebaw Fekadu**[1,3]

**1** NIHR Global Health Research Unit on Neglected Tropical Diseases, Brighton and Sussex Medical School, Falmer, United Kingdom, **2** School of Public Health, Addis Ababa University, Addis Ababa, Ethiopia, **3** Center for Innovative Drug Development and Therapeutic Trials for Africa, Addis Ababa University, Addis Ababa, Ethiopia, **4** Department of Primary Care and Public Health, Brighton and Sussex Medical School, Falmer, United Kingdom

* yirgurob@gmail.com

## Abstract

### Introduction

Scabies undermines quality of life through its highly disturbing disease symptoms, by distorting self-perception, and secondary to social stigma. Knowledge of its effect on quality of life in general and on specific aspects of day-to-day life is key to addressing the health needs of individual patients and to evaluating gains from community-based disease control interventions.

### Objectives

To measure the effect of scabies on the quality of life of people with the infestation.

### Methods

A community-based cross-sectional study was conducted in a scabies outbreak-affected district in north-western Ethiopia. The study involved 381 households and 86 adults with scabies. We used the ten-item Cardiff Dermatology Life Quality Index (DLQI) tool to collect data. Cronbach's alpha value was used to determine the internal consistency of the Amharic version of the scale. Overall and Dermatology Life Quality (DLQ) domain specific mean scores were calculated. The association between sociodemographic characteristics and scabies-related life quality impairment was tested using Kruskal-Wallis test.

### Results

Scabies moderately affected the quality of life of adults with scabies. The overall mean DLQI (mDLQI) score was 9.2 (SD = 7.6). 'Symptoms and feelings' and 'daily activity' DLQ domains had the highest mDLQI scores (3.5, SD = 1.9 and 2.2, SD = 2.5, respectively). 'Leisure activities' was the least affected domain 0.8 (SD = 1.1). In terms of severity, scabies had moderate or severe effect on DLQ of 54.7% of the participants and extremely severe

**Data Availability Statement:** All relevant data are within the manuscript.

**Funding:** GD received a financial support from the National Institute for Health and Care Research (Grant 16/136/29). The funders had no role in study design, data collection and analysis, decision to publish, or preparation of the manuscript.

**Competing interests:** The authors have declared that no competing interests exist.

effect was reported among 27% of the participants. However, no association was observed between sociodemographic characteristics and quality of life impairment.

## Conclusion

Quality of life was moderately impaired among people affected by scabies. Refocusing attention on management of disease symptoms, using standard scabies treatment, and providing psychosocial support to improve self-perception of people affected with scabies may help reduce quality of life impairment.

## Author summary

Scabies is a parasitic infestation of the outermost layer of the skin. Disease symptoms such as papular rash (mainly involving the crevices of the body), severe generalized itch and stigma attached to the infestation are believed to impair the quality of life of scabies patients. Despite the need for data, there is limited evidence about the relationship between scabies and quality of life. This study presents novel data on quality of life impairment associated with scabies and which aspects of patients' day-to-day lives are the most affected. This study was conducted in an outbreak-affected area of north-western Ethiopia where scabies had a moderate impact on the quality of life of the affected people. Disease symptoms and their psychosocial impact contributed the most to quality of life impairment. Participants in this study came from different sociodemographic backgrounds. Nonetheless, there were no major differences in DLQI scores by background characteristics.

## Introduction

The World Health Organization (WHO) defines quality of life as "an individual's perception of their position in life in the context of the culture and the value systems in which they live and in relation to their goals, expectations, standards, and concerns" [1]. This definition is further qualified, in health research, to account for the impact of diseases on health outcomes including physical, mental, and emotional wellbeing [2].

Dermatologic diseases have a serious impact on the quality of life of those affected. Bodily dissatisfaction, anxiety, and depression have been commonly reported among people affected with skin diseases [3,4]. Given the chronicity of most dermatologic conditions, the physical and psychosocial impacts are often prolonged [5]. Psychological impacts result from altered physical appearance, social stigma, and debilitating disease symptoms whose severity is determined by the size of lesions, the affected body parts, and symptom duration [5,6]. In previous research, sociodemographic characteristics such as sex, age, income, and marital status were associated with quality of life impairment [5–7].

Scabies is a parasitic skin infestation caused by the mite *Sarcoptes scabiei* [8]. Manifestations of classical scabies include severe itch and papulo-vesicular rash primarily involving body crevices such as the axillae, wrists, finger webs, buttocks, and the genitalia in men [8,9].

Scabies was recently adopted by the WHO as a neglected tropical disease [10]. Its high disease burden, skewed distribution towards under-resourced communities, and amenability to control interventions were among the criteria for the designation. This attention has revitalized the control efforts of local actors and international organizations [11].

The focus of classic scabies case management is eliminating the mite [8,12]. Manifestations of scabies infestation that undermine the quality of life of patients such as skin irritation and itch do not receive the necessary attention. Lack of evidence on scabies epidemiology and its effect on quality of life and psychosocial wellbeing of patients partly explains the narrowed focus area in scabies case management [13,14]. Data explaining quality of life impairment secondary to scabies are limited [14–16]. This study aimed to determine the overall and most affected quality of life domains among people with scabies by using community-based Dermatologic Life Quality Index (DLQI) data [17]. The findings will inform the development of a more comprehensive scabies case management and community based scabies control interventions.

## Methods

### Ethics statement

Ethical clearance was received from Addis Ababa University Institutional Review Board (IRB) and the Brighton and Sussex Medical School (BSMS) Research Governance and Ethics Committee (RGEC) with reference numbers AAUMF 03–008 and ER/BSMS9G1Z/1, respectively. Permission was sought from Amhara Regional Health Bureau and local administrative heads before the field work. Written informed consent was obtained from all participants at enrolment. A community-to-facility referral slip was issued at the end of each interview, so that people affected by scabies could get medical care from the nearest health centre.

### Study design

We conducted a population-based cross-sectional study. This survey design was chosen considering the descriptive nature of the main objective, which was to determine the effect of scabies on the quality of life of patients. The study was part of a baseline survey for a larger project that measured the off-target effect of ivermectin-based mass drug administration (MDA) for onchocerciasis elimination on scabies prevalence. The full design of the parent project is detailed elsewhere [18,19]. This article summarized aspects of the data collection and analysis that pertain to quality of life by following the STROBE reporting outline.

### Study area

We conducted the study from December 5 to 23, 2018 in Ayu Guagusa district, north-western Ethiopia, a region that had been affected by scabies outbreak since 2015 [20]. The prevalence at the time of this study was estimated at 13.4% [19]. Despite this high prevalence, only standard individual scabies care was provided at health care facilities and there were no community-based control interventions in the study district. However, a bi-annual ivermectin based MDA had been underway for onchocerciasis elimination since 2015. The district was comprised of 21 predominantly rural *kebeles* (the lowest administrative unit with an average of 500 households) [21] and all 6 study *kebeles* were rural. Three health centers provided primary health care services to the district population. Health Extension Workers (HEWs), a cadre of community health workers trained for one year, provide disease prevention and health promotion services at health posts [21,22] and *kebeles* had one health post each.

### Source population

The source population was all adult residents of Ayu Guagusa district diagnosed with scabies. Children who were screened positive for scabies in the bigger project, were referred to the nearby primary health care facilities for treatment.

### Study population

Adult members of the sample study households with clinically diagnosed scabies.

### Eligibility criteria

Adult regular members of the study households and visitors who had stayed with the family for the previous two weeks before the survey and who were diagnosed with scabies.

### Sampling

Ayu Guagusa district was purposively selected for being the only district in Agew Awi zone where ivermectin-based MDA against onchocerciasis was underway, a criterion relevant to the bigger project [19]. From the 21 *kebeles* comprising the district, six (Dekuna Dereb, Arbit, Degera, Ambera, Enavara, and Chibachibasa) were selected using simple random sampling. The *kebeles* were comprised of a varying number of *Gotes* (small villages) and one *Gote* was randomly selected from each study *kebele*. A sampling frame was prepared based on a census of households in the study *Gotes* and 381 households selected from it. All consenting adult members of the sample households that presented with scabies symptoms were examined. The sampling procedure is further explained elsewhere [19].

### Data collection procedures

Household heads or adult household members were the informants for household characteristic- related questions. Subsequently, all individual household members available at the time of the survey were interviewed and examined for manifestations suggestive of scabies. Five nurses, with ample experience caring for scabies cases, were recruited from the nearby primary hospital to collect data. A dermatovenerologist (WE) from Bahir Dar University gave them a three day refresher training before data collection. The training was meant to standardize scabies diagnosis, introduce the data collection tool, and create awareness about human subject research ethics.

Clinical diagnosis was employed to identify scabies cases. Physical examination of the study participants was conducted inside the house in a place with optimal illumination. After ensuring privacy, participants were asked to expose body parts that are typically affected by scabietic lesions such as the arms, legs, and the abdomen. However, private body parts were not involved in the examination. Other than physical examination, participants were also asked about contact history with a person presenting scabies manifestations.

### Sample size

Sample size was calculated using the formula for a continuous outcome variable in single group studies. The sample size (n = 71) was adequate to identify quality of life mean score ($\bar{x} = 10$) [14] with a standard deviation of (SD = 5.9) and margin of error 1.3.

### Data collected

The Cardiff DLQI tool was used following the guidance by Finlay and Khan [17]. The tool has been translated into 55 languages in 20 countries to measure the effect of more than 36 skin conditions, including scabies, on quality of life of patients [17,23–25]. The Amharic translation was validated in 2007 [23].

The DLQI comprises 10 items that measure patients' DLQ over a one-week recall period. Finlay and Khan recommend employing all the ten items in measuring DLQ [17]. However, two of the items were excluded in the current study. Item 6 (sports), was excluded due to the

rarity of organized sporting activities in this rural study community [26]. Item number 10 (treatment) for lack of relevance to population surveys where there was no treatment of cases involved [17]. The treatment item was originally designed to capture the effect of anti-scabietic creams, which are not convenient to apply and at times cause skin irritation, on quality of life. However, this study was conducted in a community setting which had experienced no treatment of patients. Each item had four ordinal response categories with their respective scores (i.e. Not at all = 0, a little = 1, a lot = 2, and very much = 3) [17]. The highest and lowest total scores using the original index were 30 and 0, respectively, with high scores representing more impaired quality of life. However, in this study the total score using the adapted index was 24.

## Data analysis

The data were analysed using *Stata 14.0* (StataCorp LLC, Texas) statistical software. The items were grouped to address six life quality domains that are relevant to dermatological conditions (i.e., symptoms and feelings, daily activities, work and school, leisure, personal relationship, and treatment) [17].

The total and DLQ domain-specific mean scores (mDLQI) and standard deviations were calculated. The five domains have a varying maximum possible mean scores. The 'symptoms and feelings', 'daily activity', and 'personal relationship' domains were calculated out of 6 and 'leisure' and 'work and school' domains out of 3. The difference in ranks of DLQI score by sociodemographic characteristics of participants was determined using the Kruskal-Wallis test [27].

The Hongbo *et.al.*, degrees of severity scores were applied to determine severity of impairment (i.e., no effect (0–1), small effect [2–5], moderate effect [6–10], very severe effect [11–20], and extremely severe effect [21–30]) [28]. This grading was developed based on the standard DLQI scale with a maximum score of 30, as opposed to the highest score in the current study that is 24. This difference in the two scores was adjusted by prorating the scores from each interval of severity to the standard scale [24]. The amount of missing data was very small, and we have reported this.

## Results

### Study participant profile

Eighty three households (21.7% of screened households) were found to have one or more people affected by scabies. Of the 91 adults diagnosed with scabies, 86 (94.5%) participated in this study. Data on 5 participants was dropped due to poor quality. The mean age of participants was 33 and most were between the ages 18 and 30 years. Nearly three quarters of participants had not had formal education, and only 14.1% had secondary (or above) education. Most participants (81.4%) were married. Unmarried, divorced, and widowed participants were grouped as 'single' comprising only 18.6% of the participants. The main stay of occupation was farming (87.2%) and 4.7% of the participants did not take part in any income generating activities. The distribution of households across wealth quintiles was comparable with a slight preponderance of highest wealth quintile households (32.6%) (Table 1).

### Quality of life among people with scabies

The reliability of the Amharic version scale was good, with Cronbach's α value 0.93 (>0.7). Cumulative mDLQI score among participants was 9.2 (SD = 7.6). This corresponds to moderate severity according to Hongbo [28]. 'Symptoms' and 'sexual difficulties' items had the highest (50.0%) and lowest (8.2%) proportions of participants reporting extreme quality of life

**Table 1. Sociodemographic characteristics of study participants (n = 86).**

| Variables | Categories | n (%) |
|---|---|---|
| Sex | Female | 41 (47.7%) |
| | Male | 45 (52.3%) |
| Age | 18–30 | 41 (47.7%) |
| | 31–40 | 11 (12.8%) |
| | >41 | 34 (39.5%) |
| | Mean age (SD) | 38 [15] |
| Level of education (n = 85) | No formal education | 63 (74.1%) |
| | Primary education | 10 (11.8%) |
| | Secondary or above | 12 (14.1%) |
| Occupation | Farmer | 75 (87.2%) |
| | Student | 7 (8.1%) |
| | Unemployed | 4 (4.7%) |
| Marital status | Married | 70 (81.4%) |
| | Single | 16 (18.6%) |
| Household size | < 5 people | 36 (41.9%) |
| | >5 people | 50 (58.1%) |
| Household wealth | Lowest | 13 (15.1%) |
| | Second | 18 (20.9%) |
| | Middle | 13 (15.1%) |
| | Fourth | 14 (16.3%) |
| | Highest | 28 (32.6%) |

impairment. Slightly more than half of the participants reported no effect against the items 'work and study', 'clothes', 'daily activities', and 'personal relationship' items (Table 2).

We did not observe a statistically significant difference in ranks of the total DLQI score by sociodemographic characteristics of participants or mean number of days with scabies symptoms (Table 3).

The symptoms and feelings domain had the highest mean DLQI score 3.7 $\pm$ 1.9 and leisure the lowest 0.8 $\pm$ 1.1. (Fig 1)

Scabies had severe and extremely severe effect on the quality of life of 13 (15%) and 23 (27%) participants, respectively. Mild or moderate effect was reported in 43 (50%) scabies cases. (Fig 2)

## Discussion

Quality of life was moderately impaired among people affected by scabies in north-western Ethiopia. The items: 'symptom', 'feelings', and 'work or study' were the main contributors to the overall DLQ impairment. 'Symptoms and feelings' and 'leisure activity' were the highest and lowest scoring DLQ domains. However, there was no difference in DLQI score by sociodemographic characteristics of participants or symptom duration.

In this study, we omitted the sporting activities and treatment items, from the ten that formed the standard DLQI scale, for lack of relevance to the study community. This may affect comparability of the current findings with literature. However, we assume the effect is minor as organized sporting activities were uncommon in the study area [26]. In a study among podoconiosis patients, in southern Ethiopia, the contribution of impaired sporting activities to overall impairment was marginal [25].

**Table 2. Distribution of dermatologic life quality index items among adults with scabies (n = 86).**

| Variable | Categories | Overall n (%) |
|---|---|---|
| Over the last week, how itchy, sore, painful, or stinging has your skin been (symptoms) | Very much | 43 (50.0%) |
| | A lot | 31 (36.1%) |
| | A little | 7 (8.1%) |
| | Not at all | 5 (5.8%) |
| Over the last week, how embarrassed or self-conscious have you been because of your skin (feelings) | Very much | 29 (33.7%) |
| | A lot | 13 (15.1%) |
| | A little | 7 (8.1%) |
| | Not at all | 37 (43.0%) |
| Over the last week, how much has your skin interfered with you going shopping or looking after your home or garden (work or study) | Very much | 20 (23.3%) |
| | A lot | 14 (16.3%) |
| | A little | 7 (8.1%) |
| | Not at all | 45 (52.3%) |
| Over the last week, how much has your skin influenced the clothes you wear (clothes) | Very much | 24 (27.9%) |
| | A lot | 7 (8.1%) |
| | A little | 4 (4.6%) |
| | Not at all | 51 (59.3%) |
| Over the last week, how much has your skin affected any social or leisure activities (daily activities) (n = 82) | Very much | 13 (15.9%) |
| | A lot | 11 (13.4%) |
| | A little | 12 (14.6%) |
| | Not at all | 46 (56.1%) |
| Over the last week how much has your skin been a problem at work or studying (work or study) (n = 85) | Very much | 20 (23.5%) |
| | A lot | 10 (11.8%) |
| | A little | 3 (3.5%) |
| | Not at all | 52 (61.2%) |
| Over the last week, how much has your skin created problems with your partner or any of your close friends or relatives (personal relationships) | Very much | 15 (17.4%) |
| | A lot | 10 (11.6%) |
| | A little | 13 (15.1%) |
| | Not at all | 48 (55.8%) |
| Over the last week, how much has your skin caused any sexual difficulties (sexual difficulties) (n = 85) | Very much | 7 (8.2%) |
| | A lot | 5 (5.9%) |
| | A little | 8 (9.4%) |
| | Not at all | 65 (76.5%) |

**Table 3. Demographic characteristics associated with a difference in quality of life of adults with scabies (n = 86).**

| Variables | Categories | n (%) | DLQI score mean $\pm$ SD | P-value[*] |
|---|---|---|---|---|
| Sex | Female | 47 (51.7%) | 9.2 $\pm$ 7.9 | 0.9 |
| | Male | 44 (48.4%) | 9.0 $\pm$ 7.8 | |
| Age (years) | 18–30 | 41 (47.7%) | 6.7 $\pm$ 7.0 | 0.4 |
| | 31–40 | 11 (12.8%) | 8.4 $\pm$ 7.7 | |
| | >41 | 34 (39.5%) | 10.5 + 7.9 | |
| Level of education (n = 85) | No formal education | 63 (74.1%) | 8.9 $\pm$ 7.8 | 0.9 |
| | Primary education | 10 (11.8%) | 9.9 $\pm$ 7.6 | |
| | Secondary and above | 12 (14.1%) | 9.6 $\pm$ 8.3 | |
| Occupation | Farmer | 75 (87.2%) | 9.1 $\pm$ 7.7 | 0.9 |
| | Student | 7 (8.1%) | 9.3 $\pm$ 8.9 | |
| | Unemployed | 4 (4.7%) | 8.3 $\pm$ 7.9 | |
| Marital status | Married | 70 (81.4%) | 9.4 $\pm$ 8.0 | 0.5 |
| | Single | 16 (18.6%) | 7.9 $\pm$ 6.5 | |
| Household size | < 5 people | 36 (41.9%) | 8.5 $\pm$ 7.1 | 0.5 |
| | > 5 people | 50 (58.1%) | 9.6 $\pm$ 8.2 | |
| Household wealth | Lowest | 13 (15.1%) | 12.2 $\pm$ 8.0 | 0.6 |
| | Second | 18 (20.9%) | 7.4 $\pm$ 6.4 | |
| | Middle | 13 (15.1%) | 9.2 $\pm$ 7.4 | |
| | Fourth | 14 (16.3%) | 8.9 $\pm$ 8.7 | |
| | Highest | 28 (32.6%) | 8.8 $\pm$ 8.2 | |

[*]Difference in DLQI ranks was determined using Kruskal-Wallis test.

[#] The period between the time of the first symptom and data collection.

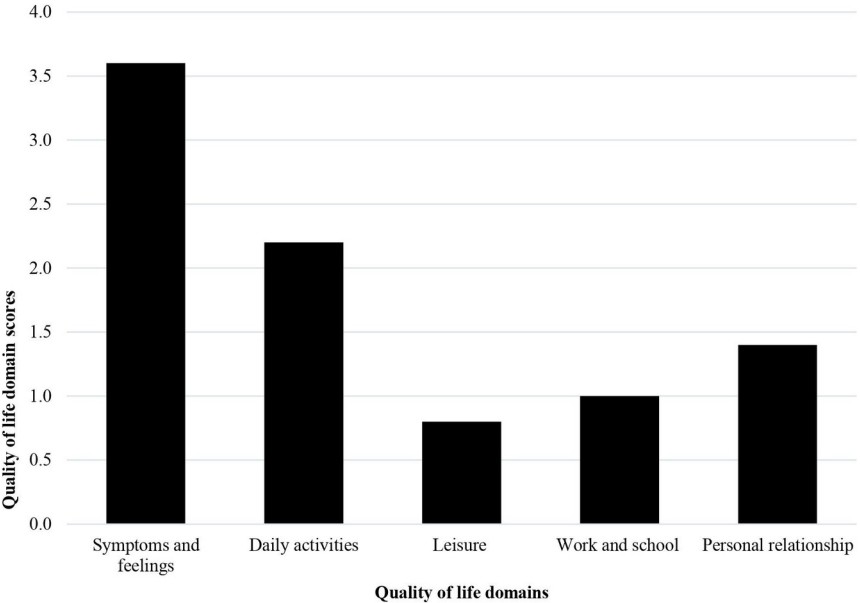

**Fig 1. Mean scores of the five quality of life domains.**

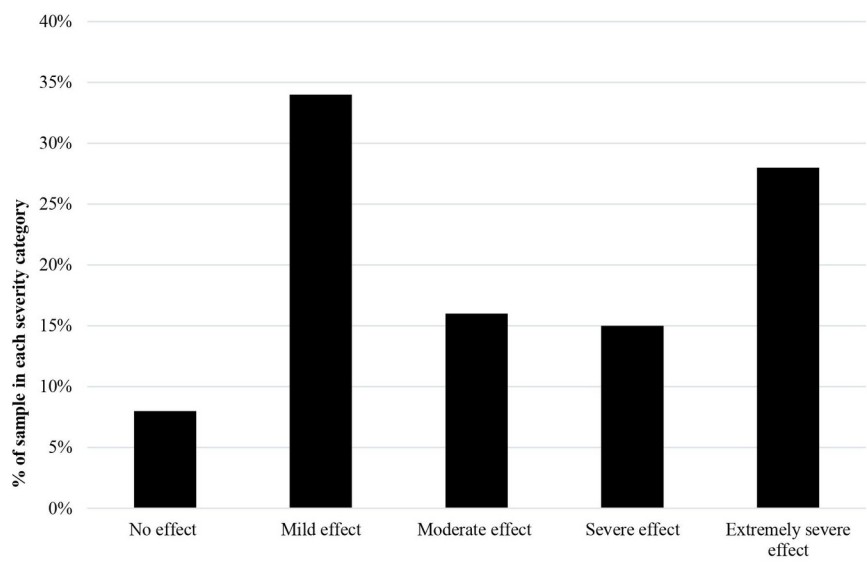

**Fig 2. Dermatology quality of life index scores by degrees of severity.**

Comparable to the current findings, studies in China and Brazil reported moderate impairment to quality of life of scabies patients [14,15]. Factors such as degree of alteration of the skin surface, body parts affected, and the type of lesion are important predictors of severity of impairment in people with dermatologic diseases [6,7]. In classical scabies, the primary lesions are papules with typical distribution [29]. Their small-size and distribution, which mainly involves parts of the body that can be covered from other people, may minimize the impact on selected life quality domains. This is also true for other parasitic infestations that commonly cause mild alterations to the skin surface such as tungiasis and pediculosis [30,31]. Conversely, other skin NTDs including podoconiosis, leprosy, and leishmaniasis, which cause severe skin lesions and bodily function impairment have severe or very severe overall impact on quality of life of patients [32–34]. Apart from presentation, disease pattern may play a part in determining its effect on quality of life. Scabies had been endemic to north-western Ethiopia and an outbreak was reported in 2015 [20], which was not controlled at the time of this study. Based on our field observation, we believe the secular trend and large-scale occurrence had contributed to a progressive reduction in the stigma attached to scabies. This may reduce the quality of life impairment by ameliorating the attributes that undermine social functioning.

The 'sexual difficulties' item contributed the least to quality of life impairment total score [14]. Sexual difficulties or dysfunction in people affected with dermatologic diseases result from physical impairment or psychosocial factors [35]. Pain and discomfort from the skin lesions and distorted self-perception bridge skin diseases and their effect on sexual function [36–38]. Classical scabies causes small-sized papular lesions with limited alterations to the skin surface [29]. This may minimize its impact on sexual appeal and performance before and during sexual intercourse. Social norms may also explain the limited impairment to sexual function. In most Ethiopian cultures discussing sex is a taboo, more so among married couples [39]. In our study, more than 80% of the participants were married. It is likely that most were hesitant to discuss any change in their sexual life with their partners.

Of the five DLQI domains investigated in this study, the highest scores arose in the 'symptoms and feelings' domain as was the case in the studies from China and Nepal [15–17]. This

domain is comprised of items measuring adverse skin sensations caused by the infestation and patients' affective response to the disease manifestations. This includes itch and irritation caused by the primary lesions, or pain and soreness that are often associated with secondary bacterial infections [40]. The intense generalized itch [41] and associated sleep disturbance may also feed into the contribution of this domain to the quality of life total score [42,43]. Optimizing symptom management is vital as getting relief from the debilitating symptoms is an important driver of care seeking for scabies [44].

The second item within the symptoms and feelings DLQ domain captures affective response of scabies patients to the disease symptoms. Skin lesions and related alterations of the skin surface undermine the mental well-being of patients causing low self-esteem, social isolation, and at times depression [15,45,46]. These responses become more intense when social stigma is attached to the disease or its symptoms [45]. Scabies lesions, which involve the hands, and the constant generalized itch, predispose scabies patients to stigma from others contributing to the negative affective response [47,48]. A study among scabies free participants of the same community reported that more than three quarters of participants preferred to avoid scabies patients [19]. The commonest responses to the anticipation of developing the infestation were fear and shame [19]. This pervasive stigma can have a toll on the mental health of scabies patients warranting psychosocial interventions alongside clinical care.

'Daily activity' was the second most affected domain. It measures the impact on participants' clothing choices and on work and study. Worth *et.al.* reported a comparable severity of impact where scabies patients living in the slums of northeast Brazil had to change into clothing which covered most parts of their body [15]. Trousers, long skirts and tops with long sleeves were preferred as they cover most of the lesions and help to prevent potential stigma [49]. The impact on clothing is more pronounced in hot weather where people prefer summer dresses to acclimate [15]. Our study population is accustomed to modest clothing with sleeves covering most parts of the body. The impairment to work and study could have resulted from the inconvenience caused by the lesions to physical activity. Sleep disturbance secondary to the itch, which worsens at night, also undermines productivity during daytime [29].

The 'personal relationship' domain was the third most affected. Of the two items in this domain the 'relationship with family and friends' was more impaired. In many communities, scabies is associated with poverty and lack of hygiene [43,49]. Such misconceptions may fuel social stigma against scabies patients. An article from this same community reported that 37% of the sample preferred to avoid scabies patients [19]. These circumstances may lead to patients limiting social relations in fear of rejection. The other reason for limiting social interactions may be to prevent transmission. Patients who are aware of the increased risk of transmission with increasing physical proximity may isolate themselves to protect people who are close to them [49]. The sociocultural context is in the background of the dynamic relationship between the disease and its effect on quality of life. Further qualitative studies should be conducted to investigate the role of community perception and social norms in mediating the impact of scabies on quality of life of patients.

Sociodemographic characteristics and duration of disease symptoms were not associated with quality of life impairment. In another study, gender shaped the effect of scabies on the quality of life of patients [15]. Females were more affected than males and negative self-perception, caused by alterations to the skin, contributed to the impairment [15]. The traditional clothing in our study population, which covers most parts of the body, may limit the severity of impairment by minimizing the effect on clothing preferences, daily activity, and stigma from others. Though we did not find a statistically significant association, the link between sociodemographic characteristics and quality of life impairment among people with scabies

needs further research. We suggest multicentre studies with larger sample sizes to investigate potential differences between the different demographic groups with more accuracy.

Longer symptom duration is an important predictor of quality of life impairment in other skin NTDs such as leishmaniasis. It correlates with severity and signifies possible complications. In the case of scabies, prolonged symptom duration may not necessarily correlate with severity. Earlier scabies research indicated a decrease in the number of mites with an increase in the duration of the infestation [50]. However, this is true only if there is neither secondary bacterial infection nor progression to crusted scabies.

The overall effect of scabies on the quality of life of the sample adult study participants was moderate, though a quarter experienced an extremely severe effect. Disease symptoms and self-perception items contributed the most to the overall quality of life impairment. Interventions that aim to improve quality of life, alongside the standard treatment, need to include management of disease symptoms. Alleviating stigma through community-based educational interventions can help reduce the impact on psychosocial wellbeing of patients.

## Acknowledgments

We would like to thank Gimja Bet hospital staff for conducting clinical diagnosis of scabies cases and data collection. Awi Zone and Ayu Guagusa District Health and administration offices for providing administrative support during field work. We are grateful to the study participants for their time.

## Author Contributions

**Conceptualization:** Jo Middleton, Jackie A. Cassell, Gail Davey, Abebaw Fekadu.

**Formal analysis:** Robel Yirgu, Jo Middleton, Stephen Bremner.

**Funding acquisition:** Gail Davey.

**Investigation:** Robel Yirgu.

**Methodology:** Robel Yirgu, Jo Middleton, Jackie A. Cassell, Stephen Bremner, Gail Davey, Abebaw Fekadu.

**Supervision:** Jo Middleton, Jackie A. Cassell, Gail Davey, Abebaw Fekadu.

**Visualization:** Jo Middleton.

**Writing – original draft:** Robel Yirgu.

**Writing – review & editing:** Robel Yirgu, Jo Middleton, Stephen Bremner, Gail Davey, Abebaw Fekadu.

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
