## [Decision Letter · Decision Letter 0]

17 Dec 2023

Dear Dr. Yirgu,

Thank you very much for submitting your manuscript "Quality of life among adults with scabies: a community-based cross-sectional study in north-western Ethiopia" for consideration at PLOS Neglected Tropical Diseases. As with all papers reviewed by the journal, your manuscript was reviewed by members of the editorial board and by several independent reviewers. In light of the reviews (below this email), we would like to invite the resubmission of a significantly-revised version that takes into account the reviewers' comments.

We cannot make any decision about publication until we have seen the revised manuscript and your response to the reviewers' comments. Your revised manuscript is also likely to be sent to reviewers for further evaluation.

Sincerely,

Aysegul Taylan Ozkan, M.D., Ph.D.,

Academic Editor

Victoria Brookes

Section Editor

Reviewer's Responses to Questions

**Key Review Criteria Required for Acceptance?**

**Methods**

-Are the objectives of the study clearly articulated with a clear testable hypothesis stated?

-Is the study design appropriate to address the stated objectives?

-Is the population clearly described and appropriate for the hypothesis being tested?

-Is the sample size sufficient to ensure adequate power to address the hypothesis being tested?

-Were correct statistical analysis used to support conclusions?

-Are there concerns about ethical or regulatory requirements being met?

Reviewer #1: The study context is confusing; 

line "32" outbreak affected ....since 2015, if the study is conducted in a community where there is uncontrolled outbreak .... this by itself has Psychological implication and bias the DLQi measurement. How could the authors explain this?

Reviewer #2: Study objectives are clearly stated at the end of the introduction, a clear testable hypothesis is not essential on such a study.

In the methods section, the study design is not clearly stated. In the study area section: this study occurred during a scabies outbreak? What percentage of the population was affected? Had people already received MDA ivermectin? Is this a very rural community? A sampling procedure was done to select households in the study area, but then were all household members examined?

The sample size was calculated to test a hypothesis? Reference number 20 in this section refers to clinical studies on not studies related to QoL and not sure how reference number 12 is helping in calculating the sample size. Please revise this section carefully.

Data collection was done by whom? Was there any missing data? How was this handled? Who diagnosed the patients with scabies? was there a field case-definition used?

Data collected: Ref #23 does not explain that sporting activities are rare in this rural study community! With regards to treatment was ivermectin MDA given to these participants? Is traditional treatment not tried in this community? You need a reference for the adapting the cumulative score and prorating the scores (lines 146-147)

I have no ethical concerns as approval for the study was obtained.

Reviewer #3: While the sampling procedure is mentioned, it lacks a detailed explanation of how the random selection of households was conducted. Providing a step-by-step description or referencing the specific publication where it is explained would be beneficial for transparency and reproducibility.

The exclusion of two items from the standard DLQI scale is mentioned, but the rationale for their exclusion is not discussed in detail. Providing more information on why these items were considered irrelevant could add transparency to the study's methodology.

**Results**

-Does the analysis presented match the analysis plan?

-Are the results clearly and completely presented?

-Are the figures (Tables, Images) of sufficient quality for clarity?

Reviewer #1: the result is well presented, no comment

Reviewer #2: The results match the data analysis plan given. Can the authors describe data handling? How was it collected, entered and analyzed using which programme?

Line 178 - correct mis-spelling "closes".

The results are clearly presented, but I have some questions:

- why was age categorization done this way ?

- lines 204-205 : better state "we did not observe statistically significant difference..." and maybe comments on some of the differences seen; for example participants in smaller households scored higher.

- in Table 3, the variable " symptom duration" is not described and difficult to understand meaning of result.

- Fig 1 , comparing mean of QoL domains this way may not be very clear. Some domains are scored out of 6, and others out of 3. The percentage of domain scores could be calculated from the total possible score for each domain and a comparatively more accurate figure obtained.

- Fig 2 , 27% of participants reported that their QoL was extremely severely affected . This important result is not brought out in your conclusions.

Can the authors comment on how many of the participants diagnosed with scabies and interviewed came form the same household in which others were also affected?

Reviewer #3: The data analysis section is concise, but it would be beneficial to include more details on how missing data, if any, was handled and whether any statistical software was used. Additionally, specifying the statistical tests used for comparisons would enhance the transparency of the analysis.

**Conclusions**

-Are the conclusions supported by the data presented?

-Are the limitations of analysis clearly described?

-Do the authors discuss how these data can be helpful to advance our understanding of the topic under study?

-Is public health relevance addressed?

Reviewer #1: the authors concluded.... timely holistic care.... a bit farfetched and exaggerated

1. time factor was not found to be significantly in your study

2. would be good to be explicit what does holistic care entitle give the moderate DLQ implication of Scabies

Reviewer #2: This section needs careful revision to relate/ compare the results of this study to findings in other scabies QoL study and QOL in other dermatological conditions in Ethiopia for each paragraph. For example paragraph starting at line 290 on daily activities: how is this related to the present study participants?

One important limitation was that the severity of scabies in participant was not recorded or the body parts affected (on line 287 - there is a comments about hands...)

Line 307 - "male sex is a strong predictor of QoL" - this statement needs clarification

The conclusions on lines 311-313 do not seem to be linked to your findings: self -perception and social functioning were the least affected QoL scores in your findings. Which finding reflect the stigmatizing effect of scabies in this group of participants?

Reviewer #3: The discussion could benefit from a more explicit comparison with similar studies conducted in other regions or countries. This would provide a broader context for understanding whether the impact on quality of life in northern Ethiopia is consistent with or differs from findings in diverse cultural and geographical settings.

The discussion mentions that sociodemographic characteristics were not associated with quality of life impairment. However, it could be valuable to elaborate on why certain sociodemographic factors were expected to have an impact and explore potential reasons for the lack of association in this specific context.

The discussion mentions stigma and social isolation as significant contributors to quality of life impairment. Providing more in-depth insights into the nature and extent of stigma, as well as potential interventions to address it, would contribute to a more nuanced understanding.

**Editorial and Data Presentation Modifications?**

Reviewer #1: (No Response)

Reviewer #2: (No Response)

Reviewer #3: (No Response)

**Summary and General Comments**

Reviewer #1: the authors concluded.... timely holistic care....

1. time factor was not found to be significantly in your study

2. would be good to be explicit what does holistic care entitle give the moderate DLQ implication of Scabies

Line 24; Scabies undermines ……….. there were only three studies, and reported moderate impact on DLQ…. …. thus is a bit over toned suggest revision. 

Line 70; complications of scabies….. left unattended……. suggest to be explicit which complications and give references. 

Line 17 to 72; lack of evidence …..narrowed case management. If there were known complications which entailed management modifications then what made it narrow? Would these complications also have stake in quality of life of affected individuals. 

Line 165; just for curiosity “unemployed” from the context of the study population……. Living on subsistence mixed farming what does it imply. 

Lines 259-266; the references cited indicated that degree of alteration, body part affected, ……. are associated with degree of DLQ ….. but in your study such variables were not considered; thus could it be arguable that you under/over estimated the implication of scabies as important variables were not included in your study. 

Line 273; sexuality: no reference is given to your claim of “taboo”. Also in subsequent paragraphs; from line 299 to 305….. in discussing the implication of scabies in personal relationships …… risky sexual behavior is indicated. In your result it was not seen as such, any reason. 

In general the manuscript needs a langue editing, before consideration

Reviewer #2: The manuscript is well written and presented and addresses a topic that is very much needed. 

The lack of comparison with individuals in the same community without the diagnosis of scabies impairs the interpretation of the results. It would be interesting to include such a control group in order to conclude how scabies affects people, or at least compare your results to other skin diseases in Ethiopia.

This being a cross-sectional study, have the authors consider completing a STROBE checklist and adding as an appendix?

In the abstract, best to mention how many participants responded (n=86) rather than how many were approached (91)

Ref #44 is incomplete

Reviewer #3: (No Response)

PLOS authors have the option to publish the peer review history of their article (what does this mean?). If published, this will include your full peer review and any attached files.

Reviewer #1: Yes: Endalamaw Gadisa Belachew

Reviewer #2: No

Reviewer #3: No
---

## [Decision Letter · Decision Letter 1]

26 Mar 2024

Dear Dr. Yirgu,

Thank you very much for submitting your manuscript "Quality of life among adults with scabies: a community-based cross-sectional study in north-western Ethiopia" for consideration at PLOS Neglected Tropical Diseases. As with all papers reviewed by the journal, your manuscript was reviewed by members of the editorial board and by several independent reviewers. In light of the reviews (below this email), we would like to invite the resubmission of a significantly-revised version that takes into account the reviewers' comments. 

We cannot make any decision about publication until we have seen the revised manuscript and your response to the reviewers' comments. Your revised manuscript is also likely to be sent to reviewers for further evaluation.

Sincerely,

Aysegul Taylan Ozkan, M.D., Ph.D.,

Academic Editor

Victoria Brookes

Section Editor

Editor comment:

As well as addressing the scientific concerns below, please review and improve the the language and structure of this manuscript as suggested by reviewers.

Reviewer's Responses to Questions

**Key Review Criteria Required for Acceptance?**

**Methods**

-Are the objectives of the study clearly articulated with a clear testable hypothesis stated?

-Is the study design appropriate to address the stated objectives?

-Is the population clearly described and appropriate for the hypothesis being tested?

-Is the sample size sufficient to ensure adequate power to address the hypothesis being tested?

-Were correct statistical analysis used to support conclusions?

-Are there concerns about ethical or regulatory requirements being met?

Reviewer #1: The study is well designed and no ethical concern, they had clear objective

Reviewer #2: Please see comments on attached pdf 

I understand that that your study covers most of the areas requested in the STORBE list, at least mention that in the methodology

Much improved paper. Thank you.

Reviewer #3: (No Response)

**Results**

-Does the analysis presented match the analysis plan?

-Are the results clearly and completely presented?

-Are the figures (Tables, Images) of sufficient quality for clarity?

Reviewer #1: The result is adequately described,

Reviewer #2: (No Response)

Reviewer #3: (No Response)

**Conclusions**

-Are the conclusions supported by the data presented?

-Are the limitations of analysis clearly described?

-Do the authors discuss how these data can be helpful to advance our understanding of the topic under study?

-Is public health relevance addressed?

Reviewer #1: The conclusion needs to be reconsidered, see my general comments.

Reviewer #2: (No Response)

Reviewer #3: (No Response)

**Editorial and Data Presentation Modifications?**

Reviewer #1: (No Response)

Reviewer #2: (No Response)

Reviewer #3: (No Response)

**Summary and General Comments**

Reviewer #1: General comments;

The authors addressed an important issue of public health interest that Transends health challenge. Thus, the result is worth communicating but

1. It needs a general language edition 

2. The authors need to stick to a line of argument in discussing their result avoiding speculations. Be consistent and see their result and context in its entirety 

Specific comments

Authors summary;

The authors said nothing about their own finding and the context under which this particular work was done. Actually, the Authors summary is to present a concise and non-technical summary of own findings in way understandable for a layperson. 

Line 48-49. The authors claimed that ………. And health education…… is needed to remove “stigma” related to scabies. To my understanding there is not result of theirs that indicated that Scabies actually causes “stigma” and the claim to remove……..perhaps could be due to choice of wording….. in any case such a problem “QoL” or “stigma” could be reduced but complete healing might not be achievable. 

Line 79-80……. The authors stated that Scabies is amenable to control interventions. Subsequently, presented …..line 98-100…… there was a biannual Ivermectin intervention in the community. But the outbreak is ongoing?? ……. This has two implications for me 1) MDA is a community-based intervention and no sign of abetting the outbreak! 2) the study was conducted under the context when majority of the community members had the disease…..line 290-293….. the authors claimed that such context might have ameliorated the attribute that undermines social functioning! Whereas in line 318…… phrases read as follows…. In a community where more than three quarters prefer avoiding scabies patients……. Related to fear and shame……. To substantiate their finding of high impact in the Symptom and feeling DLQI domain. How could the authors explain own contradictions?

Line 294…. Explaining why less effect on “sex and sexuality”……the authors claim absence/less physical or psychosocial impact…… small lesion with limited physical alteration. Whereas, discussing the work and study ….. the phrase reads as …..”the result of inconvenience …..lesions to physical activities. In addition, from line 332…. It reads …. Scabies associated with poverty and hygiene fuels stigma…. Related to fear of transmission …….. limit social interaction…… they claimed though intimate physical contact is implicated for more transmission. Then how is it possible that there claimed “taboo” and the aforementioned explain the context around sex and sexuality.

Reviewer #2: Please see comments on attached pdf 

I understand that that your study covers most of the areas requested in the STORBE list, at least mention that in the methodology

Reviewer #3: Comment 1: While the introduction provides a definition of quality of life and briefly mentions its relevance in health research, it fails to provide a broader context for the significance of studying quality of life in the context of dermatologic diseases. Readers would benefit from a more detailed discussion of the impact of dermatologic diseases on patients' overall well-being and functioning, as well as the importance of addressing quality of life issues in clinical practice and research.

Comment 2: The introduction briefly mentions that the study aims to assess the effect of scabies on quality of life using the Dermatology Life Quality Index (DLQI) data. However, it could benefit from a clearer and more explicit statement of the research objectives, including specific research questions or hypotheses that the study seeks to address.

Comment 3: The study area section lacks clear organization and structure, making it difficult for readers to follow the study design and data collection procedures. Breaking down the text into subsections such as Study Design, Sampling Procedure, Data Collection, and Data Analysis would enhance readability and comprehension.

Comment 4: The study design is briefly mentioned as a population-based cross-sectional study, however, there is limited detail provided about the rationale behind choosing this design and how it aligns with the research objectives. A clearer explanation of why this design was chosen and how it contributes to answering the research questions is necessary.

Comment 5: The description of data collection procedures lacks specificity, particularly regarding the training of data collectors and the process of clinical examination. Providing more detail on the training received by data collectors, as well as the methods used for clinical examination and diagnosis of scabies, would enhance the credibility of the study findings.

Comment 6: The authors briefly mention that there were no differences in the ranks of DLQI scores by sociodemographic characteristics or symptom duration. However, they do not delve into a deeper analysis of these factors or discuss potential reasons for the lack of associations. This represents a missed opportunity to explore important determinants of quality of life in scabies patients and may weaken the overall discussion.

Comment 7: The discussion could benefit from a section outlining potential future research directions and implications for clinical practice. This could include suggestions for further investigating the relationship between sociodemographic factors and quality of life in scabies patients, exploring interventions to address stigma and psychosocial impacts, and evaluating the effectiveness of integrated care approaches in improving patient outcomes.

PLOS authors have the option to publish the peer review history of their article (what does this mean?). If published, this will include your full peer review and any attached files.

Reviewer #1: Yes: Endalamaw Gadisa Belachew

Reviewer #2: Yes: Saba Lambert

Reviewer #3: No
---

## [Decision Letter · Decision Letter 2]

17 Jun 2024

Dear Dr. Yirgu,

Thank you very much for submitting your manuscript "Quality of life among adults with scabies: a community-based cross-sectional study in north-western Ethiopia" for consideration at PLOS Neglected Tropical Diseases. As with all papers reviewed by the journal, your manuscript was reviewed by members of the editorial board and by several independent reviewers. In light of the reviews (below this email), we would like to invite the resubmission of a significantly-revised version that takes into account the reviewers' comments. 

Please address the language requirements highlighted by one reviewer.

We cannot make any decision about publication until we have seen the revised manuscript and your response to the reviewers' comments. Your revised manuscript is also likely to be sent to reviewers for further evaluation.

Sincerely,

Aysegul Taylan Ozkan, M.D., Ph.D.,

Academic Editor

Victoria Brookes

Section Editor

Reviewer's Responses to Questions

**Key Review Criteria Required for Acceptance?**

**Methods**

-Are the objectives of the study clearly articulated with a clear testable hypothesis stated?

-Is the study design appropriate to address the stated objectives?

-Is the population clearly described and appropriate for the hypothesis being tested?

-Is the sample size sufficient to ensure adequate power to address the hypothesis being tested?

-Were correct statistical analysis used to support conclusions?

-Are there concerns about ethical or regulatory requirements being met?

Reviewer #1: the method section is okay with clear study approaches and specific methods

Reviewer #2: (No Response)

Reviewer #3: YES

**Results**

-Does the analysis presented match the analysis plan?

-Are the results clearly and completely presented?

-Are the figures (Tables, Images) of sufficient quality for clarity?

Reviewer #1: The analysis, and presentation of the results is okay

Reviewer #2: (No Response)

Reviewer #3: YES

**Conclusions**

-Are the conclusions supported by the data presented?

-Are the limitations of analysis clearly described?

-Do the authors discuss how these data can be helpful to advance our understanding of the topic under study?

-Is public health relevance addressed?

Reviewer #1: It is okay, within the scope of their results/findings

Reviewer #2: (No Response)

Reviewer #3: YES

**Editorial and Data Presentation Modifications?**

Reviewer #1: Overall the manuscript has findings that worth communication, and it is of public health interest. Yet, the language and the cascade of arguments in the discussion section need extensive edition.

Reviewer #2: (No Response)

Reviewer #3: (No Response)

**Summary and General Comments**

Reviewer #1: From the context of the study area, the public health implication of the findings and in terms of giving insight to further studies in the same line, the authors need to give due time to edit and get the manuscript to the intended audience.

Reviewer #2: This revised version is very good and ready to be accepted

Reviewer #3: I am pleased to report that all the comments and suggestions provided during the review process have been thoroughly addressed. The authors have made the necessary revisions to enhance the clarity, depth, and overall quality of the manuscript. Given the substantial improvements and the rigorous efforts undertaken to refine the paper, I am confident in recommending it for publication. The paper now meets the high standards required for publication, and I believe it will make a valuable contribution to the field. Therefore, I suggest that the paper be published.

PLOS authors have the option to publish the peer review history of their article (what does this mean?). If published, this will include your full peer review and any attached files.

Reviewer #1: No

Reviewer #2: Yes: Saba M Lambert

Reviewer #3: No
---

## [Editor Report · Decision Letter 3]

4 Aug 2024

Dear Dr. Yirgu,

We are pleased to inform you that your manuscript 'Quality of life among adults with scabies: a community-based cross-sectional study in north-western Ethiopia' has been provisionally accepted for publication in PLOS Neglected Tropical Diseases.

Best regards,

Aysegul Taylan Ozkan, M.D., Ph.D.,

Academic Editor

Victoria Brookes

Section Editor

---

## [Editor Report · Acceptance letter]

15 Aug 2024

Dear Dr. Yirgu,

We are delighted to inform you that your manuscript, "Quality of life among adults with scabies: a community-based cross-sectional study in north-western Ethiopia," has been formally accepted for publication in PLOS Neglected Tropical Diseases.

Best regards,

Shaden Kamhawi

co-Editor-in-Chief

Paul Brindley

co-Editor-in-Chief
